# Increasing Silver Nanowire Network Stability through Small Molecule Passivation

**DOI:** 10.3390/nano9060899

**Published:** 2019-06-20

**Authors:** Alexandra Madeira, Marie Plissonneau, Laurent Servant, Irene A. Goldthorpe, Mona Tréguer-Delapierre

**Affiliations:** 1CNRS, Institut de Chimie de la Matière Condensée de Bordeaux, University Bordeaux, UMR 5026, 33687 Pessac, France; madairalexandra@yahoo.fr(A.M.); marie.plissonneau@solvay.com (M.P.); 2Institut des Sciences Moléculaires, University of Bordeaux, UMR 5255 33405 TALENCE CEDEX, France; Laurent.servant@u-bordeaux.fr; 3Department of Electrical & Computer Engineering and The Waterloo Institute for Nanotechnology, University of Waterloo, Waterloo, ON N2L 3G1, Canada; igoldthorpe@uwaterloo.ca

**Keywords:** silver, nanowire, passivation, 11-mercaptoundecanoic acid

## Abstract

Silver nanowire (AgNW) transparent electrodes show promise as an alternative to indium tin oxide (ITO). However, these nanowire electrodes degrade in air, leading to significant resistance increases. We show that passivating the nanowire surfaces with small organic molecules of 11-mercaptoundecanoic acid (MUA) does not affect electrode transparency contrary to typical passivation films, and is inexpensive and simple to deposit. The sheet resistance of a 32 nm diameter silver nanowire network coated with MUA increases by only 12% over 120 days when exposed to atmospheric conditions but kept in the dark. The increase is larger when exposed to daylight (588%), but is still nearly two orders of magnitude lower than the resistance increase of unpassivated networks. The difference between the experiments performed under daylight versus the dark exemplifies the importance of testing passivation materials under light exposure.

## 1. Introduction

Ag nanowire networks have numerous technologically relevant properties that have been explored for use in strain sensors, transparent flexible conductors, light emitting diodes, e-paper, self-healing electronic devices, liquids crystal displays, artificial skins, and solar cells [1]. Their application as transparent electrodes in particular, has received wide attention in the past several years [2,3]. Commercially used transparent electrodes are typically films of metal oxides such as indium tin oxide (ITO), however, these metal oxides have limited mechanical flexibility and require high temperatures and vacuum for deposition. Nanowire (NW) networks have emerged as an attractive replacement since solution-synthesized Ag nanowires can be mass produced, and can be easily deposited on a surface using roll-to-roll industrial processes at room temperature [4,5,6,7,8]. They are also mechanically flexible and can have conductivity and transparency values as high as metal-oxide based transparent electrodes [1]. However, one of the outstanding problems of these nanowire films is their chemical instability [9,10,11,12].

Ag is prone to oxidation and sulfidation when exposed to air, resulting in a deterioration of electrical properties [9,13,14,15,16]. The metal corrodes due to chemical reactions between its surface and the hydrogen sulfide, water, or carbonyl sulfide (OCS) in air. Analyses of the NW surfaces performed by Elechiguerra [9], Deignan [12], and Chen [17] all show the main corrosion artifact to be Ag_2_S in the form of nanoparticles or a discontinuous shell. This corrosion has adverse effects on the electrical properties of AgNW electrodes. For example, Moon et al. have observed a 240% increase in the sheet resistance of electrodes based on poly(vinylpyrrolidone) (PVP)-stabilized Ag nanowires after two months of air-storage [18]. Electrodes made of thinner AgNW@PVP (25 nm in diameter) were non-conductive after only a couple of weeks [12]. 

To enhance the chemical stability of AgNW networks in air, they can be coated with thin protective layers. However, for many applications, a suitable passivation layer is lacking due to the stringent requirements. The passivation layer must be optically transparent, inexpensive, easy to deposit, and mechanically flexible so that the advantages of the silver nanowire electrode can be fully realized. In addition, in devices such as solar cells and light emitting diodes (LEDs), the nanowire passivation layer must also permit the flow of current from the electrode into or out of the device. 

Several AgNW transparent electrode passivation materials with many of the above properties, most commonly conductive polymers, graphene, and thin metal oxide layers, have been studied [19,20,21,22,23]. The performance of PEDOT:PSS (poly(3,4-ethylenedioxythiophene)-poly(styrenesulfonate)) as a passivation material is insufficient, with the sheet resistance of the silver nanowire electrode increasing nearly 9% in less than five days of exposure to air [24,25]. Protection with graphene flakes is similarly unsuitable, with resistance increasing by almost 50% over eight days at 70 °C [26]. Layer(s) of graphene deposited by chemical vapor deposition are more effective [27]. Impermeable to numerous gas molecules, they effectively protect the metal from harmful gases such as hydogen sulfide or carbonyl sulfide. However, their potential integration into industrial devices is limited because the deposition of one layer of graphene onto metal networks is difficult and expensive, and subsequent functionalization is not straightforward. Coating the AgNW networks with thin oxide layers is also frequently used to achieve both stability and robustness of the transparent electrodes. For example, Song et al. showed the enhanced chemical stability of titania coated AgNW electrodes [19]. However, a degradation of the mechanical properties of the hybrid electrodes was observed since the titania did not bond well to the underlying substrate. Moreover, all these passivation layers reduce the transparency of the electrodes. PEDOT:PSS, graphene, and TiO_2_ decreased the electrode transparency by 8, 2 and 3 percentage points, respectively [25,26,27], which are significant amounts considering a transparency > 90% is typically desired for several applications. 

Instead of coating a passivating material over the entire electrode surface as in the paragraph above, coating the surfaces of the AgNWs only with an organic short molecule has many advantages. Functionalization of the metal nanowires with strongly bound ligands enhance their chemical stability, and in many cases does not affect the transparency and conductivity of an electrode [28,29,30]. It should also not have any effect on the electrode’s mechanical flexibility, and the process is cost-effective. Idier et al. evaluated the benefit of the passivating a AgNW electrode with triphenylphosphine (PPh_3_) [28]. Although the PPh_3_-protected AgNW network showed higher chemical stability to air than PVP-coated ones, there was still a 500% increase in resistance after 110 days. Liu et al. recently tested the passivation abilities of several different organothiols. For example, with 2-mercaptobenzimidazole (MBI), the resistance increase of a AgNW network was limited to only 67% after 120 days [30]. However, these latter tests were performed in a chamber without exposure of the AgNWs to light. We will show here that testing passivation materials in light is critical.

Since short molecules have many attractive qualities as a passivation strategy, it is worth seeing if other molecules can be more effective. Testing in light must also done. The short molecule 11-mercaptoundecanoic acid (MUA) can easily attach to Ag through a thiolate bond and has been widely used to modify Ag and Au surfaces [31,32,33,34]. We recently showed using surface enhanced Raman spectroscopy that the PVP remaining on AgNWs, after their synthesis, could easily be replaced by MUA molecules by a simple immersion of the NWs in a solution of the thiolated compound [35]. A MUA monolayer on Ag forms a packing order due to the electrostatic and Van der Waals forces between the chains [33], which in turn can act as a barrier against corrosion. Moreover, by saturating the surface of AgNW with sulfur functions, the formation of extra Ag_2_S could be avoided as every surface atom of silver would already be bonded to sulfur. Thus, the key advantage of the MUA molecule is its small size (~20 Å), and its ability to self-assemble onto the surface of silver in alcohol yielding to the formation of a compact and stable monolayer. Furthermore, the possibility to post-functionalize the carboxylated-terminated monolayer can be interesting for manipulating the metal surface properties (wettability, permeability, conductivity, (bio-)sensing efficiency etc.). 

Very recently, AgNWs were passivated with alkanethiolate (octadecanethiol, ODT), a similar molecule to MUA but with 18 carbon chains instead of 10, and having no acidic function at the molecule end [29,30]. An 18% resistance increase was measured after 30 days versus 169% for unpassivated nanowires. However, the nanowires were embedded in an epoxy resin and thus the passivation effectiveness of the molecule alone is unclear. Furthermore, the surface of a nanowire network cannot be covered in a non-conductive polymer if used in a solar cell or LED since the electrode is not electrically accessible. In a later study, the effectiveness of ODT was studied without embedding the nanowires in a resin, but only tests in the dark were performed whereas in applications, AgNW electrodes will absolutely be exposed to light [30].

In this work, we explore the effectiveness of MUA in reducing the corrosion tendency of AgNWs in air. In our previous paper, only a preliminary investigation of the effectiveness of MUA on the electrical stability of a Ag nanowire network was done [35]. Here, we conduct stability tests of MUA-coated AgNWs under both daylight and in the dark, compare its efficacy to a PPh_3_ coating, report results on nanowires of different diameters, and determine the effect of MUA on the transparency and initial sheet resistance of the nanowire network. We also report a more effective strategy of passivating the nanowires with MUA after their formation into electrodes.

## 2. Materials and Methods

***Chemicals.*** Silver nitrate (AgNO_3_, 99.99%), PVP (average Mw ~1,300,000 g·mol^−1^), sodium bromide (NaBr), 1,2-propanediol (ACS reagent, ≥99.5%), triphenylphosphine (PPh_3_, ≥95%) and 11-mercaptoundecanoic acid (MUA, 98%) were procured from Sigma Aldrich (Saint Louis, MO, USA), and used without further purification.

***Ag nanowire synthesis***. AgNWs were synthesized using a modified protocol based on a method recently patented [36]. This approach produces uniform silver nanowires in high yield and in a single run. First, 166 mg of PVP and 25 mL of 1,2-propanediol were mixed for 45 min at 160 °C. Then 50 µl of NaBr (50 mM) dissolved in 1,2-propanediol was added. After another 15 min, a syringe pump was used to continuously add 10 mL of AgNO_3_ (100 mM) dissolved in 1,2-propanediol, at a rate of 0.15 mL/min for 1 h. Statistical analysis TEM imaging revealed an average diameter of 32 ± 10 nm and an average length of 40 ± 22 µm. Nanowires of 70 ± 12 nm (and length of 80 ± 25 µm) were produced by decreasing the amount of PVP added (150 mg instead of 166 mg). At the end of the synthesis, the Ag nanowires have a passivation layer of PVP (~3 nm thick) that avoids aggregation in solution. 

***Nanowire network preparation and passivation***. The NW solution (5.7 mg/mL) was deposited on 24 × 32 mm glass substrates (Roth glass) by Mayer rod coating [7]. Four coats in 4 orthogonal directions were applied so that a random NW network was achieved. The samples were then annealed at 160 °C for 30 min to reduce the resistance between overlapping NWs. Silver paste and copper tape was affixed to the ends of the electrodes to allow for electrical measurements. 

Due to the strong affinity between the thiol function and silver, MUA can easily replace PVP on the wire surface [35]. To perform this surface modification, the NWs on glass were immersed in an 18 mM solution of MUA in ethanol. Continuous stirring of the MUA over the NWs was achieved by placing the containers on a rocking platform for 1 h 30 min. A similar protocol was used to passivate nanowire networks with PPh_3_ (18 mM). A simple schematic of the overall process is illustrated in the Scheme 1. In this experimental condition, the MUA fully saturated the surface. Previous Raman investigations evidenced that a saturated layer was produced within 70 min, with a MUA solution of 5 mM [35]. 

***Characterization***. Samples were imaged with scanning electron microscopy (SEM) and atomic electron microscopy. X-Ray Photoelectron spectroscopy (XPS) measurements were conducted with a Thermo Fisher Scientific K-Alpha spectrometer (Waltham, MA, USA), and spectra were fitted and analyzed with Thermo Scientific Avantage software (Waltham, MA, USA). The initial resistances of the electrodes were measured by a four-point probe measurement. Quoted resistances are the average resistance of eight electrodes prepared in the same way. For the stability data, the relative changes in the sheet resistance were instead measured by two-point probe measurements, since the repeated measurements required in the lifetime study could damage the electrode at the contact points. Transparency values were measured at 550 nm with plain glass as a reference. Initial transparency measurements were obtained with a UV-Visible SHIMADZU spectrophotometer (Kyoto, Japon), then a handheld window tint meter was used throughout the remainder of the lifetime studies. 

***Stability tests over time.*** Electrodes were stored in the ambient atmosphere of the laboratory (average relative humidity 80%). Some electrodes were exposed to daylight and the others were stored in the dark. Electrical and optical measurements were performed over a period of 4 months, with the optical measurement being done less frequently since the latter measurement caused some nanowires come off the glass.

## 3. Results

Scheme 1 depicts the route adopted to passivate the surface of PVP-coated AgNW network. 

This strategy of passivating the nanowires after their formation into electrodes, rather than before, was necessary to achieve a uniform dispersion of nanowires in the network. When the NWs were passivated with MUA in suspension, such as done in our prior work [35], there was significant aggregation when they were subsequently deposited as a film due to the weak repulsion between MUA-capped AgNWs (Appendix A). The agglomeration affected the transparency of the NW electrodes passivated before their formation into networks (Appendix A). Immersing the AgNW networks in a solution of MUA resulted in a uniform layer of MUA on the AgNWs and no MUA on the open areas of the substrate (Figure 1a and Appendix A). The MUA layer on the nanowires is so thin that atomic resolution microscopy (ARM) and elemental analysis were difficult (Figure 1b, Appendix A). 

Table 1 reports the initial sheet resistances and transparencies of AgNW networks with and without MUA passivation. The values observed are similar, indicating that the MUA molecule coating neither affects the electrical nor the optical properties of the electrodes. It also indicates that immersing the NW network in a MUA/ethanol solution does not remove NWs from the glass substrate.

Figure 2 shows the average resistance change of the unpassivated and passivated 32 nm diameter AgNW networks stored in the dark. The sheet resistance of the unpassivated samples increased by 48% over the four months, whereas that of the MUA-capped silver nanowires only increased by 12%. These results suggest that the MUA acts an effective barrier layer against corrosion. Exposure to air leads to a progressive deterioration of PVP-coated Ag nanostructures (break down near junctions as well as silver oxide formation). In contrast, the morphologies of the AgNWs protected by the MUA layer were unchanged (Appendix A). No sign of oxidation was detected. MUA forms a coordination complex with the metal, which can effectively protect it from external contaminant sources. The strong Ag–S bonds protect the surface from oxidation. 

Figure 2 also shows similar investigations for networks stored and exposed to daylight. In this case, the resistance of the unpassivated NWs increased significantly (10,960%) after four months. Under UV-visible exposure, corrosion is clearly sped up compared to the dark. Light acts as a catalyst for the oxidation process and the formation of silver oxide (Appendix A). UV light ozone dissociation may favor metal oxide formation, which is much less conductive than Ag [9,37]. Compared to the reference sample, the MUA-coated electrodes experienced a dramatically lower sheet resistance increase of 588%. This observation confirms the ability of the thin MUA layer to prevent Ag corrosion (Appendix A). The MUA layer deoxidizes the Ag surface, making the metal more chemically stable. It reduces the access of air molecules (O_2_, H_2_O, H_2_S…) to the nanowire surface by forming a stable barrier layer.

Table 2 shows the evolution of the transparency of the different samples over time. The results confirm that the MUA coating does not interfere with the transparency in neither light nor dark storage conditions. 

These overall observations are also true for thicker nanowires. Similar conclusions could be drawn from investigations performed on networks composed of 70 nm diameter AgNWs (Appendix A). The drastic effect of light on the electrical properties of both the unpassivated and passivated electrodes shows that all silver nanowire electrode passivation materials need to be tested under light, particularly since in applications these electrodes will certainly be exposed to light. 

The performance of the MUA passivation was also compared to the surface modification with PPh_3_ molecules. We tested PPh_3_ passivated electrodes in the light and measured a 772% increase of the sheet resistance over 4 months, slightly higher than the 588% increase measured with MUA-passivated electrodes (Appendix A). The small difference observed in the kinetics of silver photo-oxidation may reflect small chemical differences at the surface: variation in the ligand densities and surface oxidant states. One possibility is that when the nanowires are passivated with MUA, the thiols reduce or displace the oxide on the silver surface. When functionalizing with PPh_3_, on the other hand, oxide could be allowed to form before a full passivation layer is obtained. This oxide could accelerate silver corrosion over time. Furthermore, small differences in the level of coverage may affect the oxidation process. 

## 4. Discussion

The increase in resistance of MUA-passivated AgNW electrodes when exposed to light is still high, and approaches to improve the passivation will be needed. One possibility is to test the effectiveness of alkanethiols with longer chains, as this will result in a thicker protective barrier of carbon. Another would be to investigate the impact of the structural order of the organic molecules on the silver surface. Non-ordered layers of thiolated spacers may affect the chemical stability of the metal. Similarly, a modification of the MUA carboxyl group as well as mixed ligand layers could be investigated. For example, it would be interesting to attach benzene cycles to MUA, which can be achieved through an amide bond. Since the benzene has an important electronic density, it would allow to stabilize the silver at the zero-valence state by mesomeric effect.

## 5. Conclusions

This study shows that MUA is an easy-to-deposit passivation layer that improves the stability of silver nanowire electrodes while not affecting their transparency. It was shown that testing passivation materials under light conditions is critical, as light greatly affects electrical stability of the electrode. The sheet resistance of electrodes passivated with MUA increased by 588% after four months when exposed to daylight versus 10,960% for unpassivated electrodes. In order to improve this result, testing molecules with longer carbon chains or modifying the carboxl group of the MUA molecules is suggested. This work should be applicable for increasing the stability of a range of Ag nanostructures for various applications.

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
