# Peer review of "Increasing Silver Nanowire Network Stability through Small Molecule Passivation"

_nanomaterials, 2019, doi:10.3390/nano9060899_

Reviewer 1 Report

Maderira et al. and group presented a transparent AgNW electrode with high stability by using small organic molecules of 11-mercaptoundecannoic acid (MUA). The transparent AgNW electrode with MUA increases by only 12% over 120 days when exposed to atmospheric conditions but kept in the dark. The increase is larger  when exposed to daylight, 588%, but is still nearly two orders of magnitude lower than the resistance increase of unpassivated networks. Paper can be published on Nanomaterials after major revision. Here some comments to improve paper. 1. MUA is key component in this paper, so more detail discussion about chemical properties and fundamental  characteristics of MUA should included. 2. The AgNW  electrode with MUA at various thickness should be studied. 3. A comparison table about performance of transparent AgNW @ MUA with previous published on AgNW-based electrodes should be presented. 4. Some applications of transparent AgNW @ MUA electrode should be presented.

Author Response

1. MUA is key component in this paper, so more detail discussion about chemical properties and fundamental characteristics of MUA should included.

The reviewer is right. We have modified the text in the introduction (see p.3).

Line 110… "by saturating the surface of AgNW with sulfur functions, the formation of extra Ag2S could be avoided as every surface atom of silver would already bonded to sulfur." Thus, the key advantage of the MUA molecule is its small size (~ 20 Å), its ability to self-assemble onto the surface of silver in alcohol yielding to the formation of a compact and stable monolayer. Furthermore, the possibility to post-functionalize the carboxylated-terminated monolayer can be interesting for manipulating the metal surface properties (wettability, permeability, conductivity, (bio-)sensing efficiency, etc…).

2. The AgNW electrode with MUA at various thickness should be studied.

Longer adsorption times or an increase of the concentration of the MUA solution do not increase the thickness of the self-assembled monolayer since the standardized exposure time (1H30) and concentration (18 mM) adopted yields a saturated surface. We have re-emphasized this fact on page 4.

Further increasing the thickness would require a post-functionalization step of the molecular monolayer via the preparation of anhydride intermediates or via cross-linking of the carboxyl function to amine group by the EDC-NHC protocol. This would require large efforts to prepare the intermediates, characterize the bilayer formation by XPS investigations and test the long term stability of the electrodes. We believe that such pathways could be put forward in follow-up studies of our work, by us or others.

3. A comparison table about performance of transparent AgNW@MUA with previous published on AgNW-based electrodes should be presented.

We thank the referee for her/his suggestion. We added a table comparing the performances of the nanowires passivated before or after their formation into electrodes in the supplementary material section (Figure S1 and Table S1). Passivating the nanowires with MUA in suspension and subsequently deposited them as a film mainly affects the transparency of the electrode since the small surface ligand yields to some aggregation of the AgNWs.

4. Some applications of transparent AgNW @ MUA electrode should be presented.

Silver nanowire electrodes have been demonstrated in a large range of devices in many papers in the literature. Demonstrating a device with MUA-coated silver nanowires would be very similar to these other works. We think it’s best for the paper to focus on MUA’s passivation efficacy to keep the paper more concise. 

Reviewer 2 Report

In this manuscript, the authors reported a method of passivating silver nanowires to improve their stability. The authors demonstrated the reduced sheet resistance changes of MUA-coated silver nanowire network when exposed to daylight over 120 days. In my opinion, this passivation method can be useful for the application of the silver nanowire network to various industries including transparent electrodes. However, it seems that the experimental data to support the authors' claims are somewhat lacking. Therefore, I sincerely recommend publishing this manuscript in Nanomaterials, after major revision including the points raised below

1.     In Fig.1, it was mentioned that MUA layer is uniformly coated on the silver nanowire, not on the substrate. However, there is no experimental data to support this explanation. I recommend, if possible, to add experimental data, such as XPS or Raman analysis, to support the explanation that the MUA is only coated on the silver nanowire.

2.     On page 6, line 207 ~ 208, the author noted that unlike PVP-coated silver nanowires, the silver nanowire protected by the MUA layer were not changed in morphologies. I recommend adding SEM image to show this.

Author Response

1.     In Fig.1, it was mentioned that MUA layer is uniformly coated on the silver nanowire, not on the substrate. However, there is no experimental data to support this explanation. I recommend, if possible, to add experimental data, such as XPS or Raman analysis, to support the explanation that the MUA is only coated on the silver nanowire.

In response to this comment, we added the following data to the Supp. Info. Section. We conducted  EDX chemical analysis on different areas of the MUA-coated networks: (i) areas without AgNWs (ii) areas with AgNWs@MUA. The comparison clearly shows that no traces of MUA were found on the areas without nanowires.

We now refer to this data in our manuscript :

Immersing the AgNW networks in a solution of MUA leads to the formation of AgNWs with a thin and uniform layer of MUA and no MUA deposition on the open areas of the substrate (Figure 1(a) and Figure S2).

2.     On page 6, line 207 ~ 208, the author noted that unlike PVP-coated silver nanowires, the silver nanowire protected by the MUA layer were not changed in morphologies. I recommend adding SEM image to show this.

To clarify this point, we added a SEM image of the MUA-coated silver nanowires stored 4 months  in the Figure S5 of the Supp.Info. section.

Round  2

Reviewer 1 Report

 The manuscript has been significantly improved to publish in Nanomaterials.

Author Response

We thank the reviewer for this positive appraisal.

Reviewer 2 Report

I have confirmed that you have made the appropriate modifications to the paper I mentioned.

However, some minor errors are found and recommended to be corrected.

1.  The contents of the figure do not match the figure caption in Fig.S6 and S7 in Supplementary Section 4. In other words, Figure S6 looks like EDX data, not TEM image and XPS, and Figure S6 is also not XPS spectrum.

2. Figure S9 in Supplementary Section 5 is also not covered by the figure caption.

3. The same is for Figure S10 in Supplementary Section 6.

The author should show supplementary data with the appropriate content described in the text.

Author Response

The contents of the figure do not match the figure caption in Fig.S6 and S7 in Supplementary Section 4. In other words, Figure S6 looks like EDX data, not TEM image and XPS, and Figure S6 is also not XPS spectrum.

We do not agree with the reviewer. Figure S6 shows a TEM image of PVP-coated nanowires fabricated 4 months ago as well as a Xray photoelectron spectrum of the AgNW network.

The Figure S7 shows a Xray photoelectron spectrum of the MUA-coated AgNW network.

2. Figure S9 in Supplementary Section 5 is also not covered by the figure caption

The Figure caption does fit with the graphs reported. Both graphs shows the evolution of the sheet resistances of AgNW networks as a function of time. 

3. The same is for Figure S10 in Supplementary Section 6.

The reviewer is right and we thank him for picking up this inaccuracy. We have changed the text (see page 1 of the Supp. Mat. Section).